# National 30-Day Readmission Trends in IBD 2014–2020—Are We Aiming for Improvement?

**DOI:** 10.3390/medicina60081310

**Published:** 2024-08-13

**Authors:** Irēna Teterina, Veronika Mirzajanova, Viktorija Mokricka, Maksims Zolovs, Dins Šmits, Juris Pokrotnieks

**Affiliations:** 1Department of Pharmacology, Faculty of Pharmacy, Riga Stradiņš University, LV-1007 Riga, Latvia; 2Faculty of Medicine, Riga Stradiņš University, LV-1007 Riga, Latvia; veronika.mirzajanova@gmail.com; 3Pauls Stradiņš Clinical University Hospital, LV-1002 Riga, Latvia; viktorija.mokricka@stradini.lv (V.M.); pokrot@latnet.lv (J.P.); 4Statistics Unit, Riga Stradiņš University, LV-1007 Riga, Latvia; maksims.zolovs@rsu.lv; 5Institute of Life Sciences and Technology, Daugavpils University, LV-5401 Daugavpils, Latvia; 6Department of Public Health and Epidemiology, Faculty of Health and Sports Sciences, Riga Stradiņš University, LV-1007 Riga, Latvia; dins.smits@rsu.lv; 7Department of Internal Diseases, Faculty of Medicine, Riga Stradiņš University, LV-1038 Riga, Latvia

**Keywords:** inflammatory bowel disease, Crohn’s disease, ulcerative colitis, readmissions, trends

## Abstract

*Background:* Inflammatory bowel disease (IBD) prevalence in Eastern Europe is increasing. The 30-day readmission rate is a crucial quality metric in healthcare, reflecting the effectiveness of initial treatment and the continuity of care post-discharge; however, such parameters are rarely analyzed. The aim of this study was to explore the trends in 30-day readmissions among patients with inflammatory bowel disease in Latvia between 2014 and 2020. *Methods:* This is a retrospective trends study in IBD—ulcerative colitis and Crohn’s disease (UC and CD)—patients in Latvia between 2014 and 2020, involving all IBD patients identified in the National Health service database in the International Classification of Diseases-10 (ICD) classification (K50.X and K51.X) and having at least one prescription for IBD diagnoses. We assessed all IBD-related hospitalizations (discharge ICD codes K50X and K51X), as well as hospitalizations potentially related to IBD comorbidities. We analyzed hospitalization trends and obtained the 30 day all-cause readmission rate, disease specific readmission rate and readmission proportion for specific calendar years. Trends in readmissions and the mean length of stay (LOS) for CD and UC were calculated. *Results:* Despite a decrease in admission rates observed in 2020, the total number of readmissions for CD and UC has increased. Female patients prevailed through the study period and were significantly older than male patients in both the CD and UC groups, *p* < 0.05. We noted that there was no trend for 30 day all-cause readmission rate for CD (*p* > 0.05); however, there was a statistically significant trend for 30 day all-cause readmission for UC patients (*p*-trend = 0.018) in the period from 2014 to 2019. There was a statistically significant trend for CD-specific readmission rate (*p* < 0.05); however, no statistically significant trend was observed for UC-specific readmission (*p* > 0.05). An exploratory analysis did not reveal any statistically significant differences between treated and not-treated IBD patients (*p* > 0.05). The increasing trend is statistically significant over the period 2014–2018 (*p* < 0.05); however, the trend interrupts in 2020, which can be associated with the COVID-19 global pandemic and the related changes in admission flows where the gastroenterology capacity was reallocated to accommodate increasing numbers of COVID-19 patients. More studies are needed to evaluate the long-term impact of COVID-19 pandemic and 30-day readmissions. No significant dynamics were observed in the mean total hospitalization costs over the 2014–2020 period.

## 1. Introduction

Inflammatory Bowel Disease (IBD) is a growing global health concern, having a significant impact on healthcare systems and patients’ quality of life [1]. The disease is characterized by the chronic progressive or relapsing–remitting inflammation of the gastrointestinal tract, and its management is complex, often requiring long-term medical therapy, surgical intervention, or both [2]. IBD may affect all age groups, from pediatric to elderly patients [3]. The burden of IBD is particularly high in Europe, with increasing direct and indirect healthcare costs [1]. These trends highlight the need for sustainable and cost-effective IBD care.

Assessing the quality of inpatient care and post-discharge outpatient care often involves considering the rate of hospital readmissions. While many readmissions are unavoidable, it is crucial to address the preventable cases. Readmissions within 30 days of discharge are used as a quality metric for the care of hospitalized patients. A range of factors contribute to increased 30-day readmission rates in IBD patients, including inadequate pain control and the need for parenteral nutrition [4]. These readmissions are associated with longer hospital stays and comorbid conditions [5]. In older patients, a high degree of disability and comorbidity, as well as the diagnosis of sepsis, are independent risk factors for potentially preventable readmissions [6]. Addressing these factors is crucial, and a multifaceted approach, including behavioral, clinical, educational and psychosocial components, may be beneficial [7].

Understanding the hospitalization trends and avoidable readmission rates in the Latvian population is critical for optimizing care delivery, reducing healthcare costs and ultimately improving patient outcomes. As noted in a previous study, the overall comorbidity burden increases with time, indicating the importance of long-term management and monitoring of IBD patients [8]. Therefore, this national retrospective study was designed to explore the trends in 30-day readmissions among patients with IBD (UC and CD) in Latvia between 2014 and 2020. The study timeframe aimed to explore COVID-19 impact (inclusion of 2020) as well and compare it to the other years.

## 2. Materials and Methods

### 2.1. Design and Data Sources

This is a retrospective trends study in IBD (UC and CD) patients in Latvia between 2014 and 2020. The data were extracted from Latvian Center of Disease Prophylaxis and Control (CDC), who is the data holder for the National Health Service (NHS). The Latvian healthcare system provides universal health coverage, including the majority of elective and all emergency hospitalizations, meaning the study covers almost all hospitalization cases for IBD in Latvia. The data are collected using International Classification of Diseases [9], Tenth Revision and Clinical Modification (ICD-10-CM/PCS) codes. This research used the anonymized dataset provided by CDC, where hospitalization and reimbursed medicines data were linked based on an encryption-generated single time unique patient identifier. Therefore, the dataset did not contain any personal identifiable information.

### 2.2. Study Population

The study includes all 30-day readmissions of CD (ICD code K50.X) and UC (ICD code K51.X) patients identified as IBD patients via the prescription medicines database and had at least one filled in prescription for IBD. Elective, traumatic and daycare hospitalizations were excluded from the study. Patient transportation cases between hospitals and departments were recorded as one admission case based on the first department and/or hospital that admitted the patient.

We assessed all the IBD-related hospitalizations (hospitalization discharge ICD codes K50.X and K51.X, referred as UC-specific and CD-specific readmission), as well as hospitalizations potentially related to IBD comorbidities using discharge codes: infectious and parasitic diseases (ICD codes—A00-B99), diseases of the digestive system (K00-K93), neoplasms (C00-D48), and diseases of the musculoskeletal system and connective tissue (M00-M99). Total “all-cause” readmission calculation includes all readmissions under ICD codes A, K, C, M. Readmissions were calculated for every year from 2014 to 2020. Using unique hospitalization identifiers, the first hospitalization of CD and UC patients was identified and one subsequent hospitalization within 30 days was tagged as a readmission.

### 2.3. Statistical Analysis and Outcome Measures

The data were analyzed using Jamovi program (Version 2.3) and Joinpoint regression program (Version 5.0.2). The *p* value ≤ 0.05 was set as the threshold for statistical significance. We highlighted hospitalization trends and obtained the 30 day all-cause readmission rate, disease specific readmission rate and readmission proportion for specific calendar years. Trends in readmissions and mean length of stay (LOS) for CD and UC were calculated. Hospitalization costs were analyzed using hospital day cost based on hospital status using Republic of Latvia Cabinet of Ministers regulation Nr. 555 (“Financing of Healthcare”), 2023 version [10]. Hospitalization costs cover hospital stay; the procedure, manipulation and other hospitalization-associated costs are excluded from the calculation, due to a high degree of variability. A special type of regression called “joinpoint” was used to analyze trends in data over time, particularly when those trends might change at specific points [11]. An exploratory analysis was conducted to compare trends in 30-day readmission in treated vs. untreated IBD patients. A treated patient was defined as having filled in at least 5 prescriptions for IBD before the hospitalization. The data distribution was assessed via inspection of the normal Q-Q plot and with the Shapiro–Wilk test. Data homogeneity was assessed with the Levene test. An independent samples *t*-test was used to compare age and rate of diseases between genders. The study has received Rīga Stradiņš University Ethics committee approval (27 December 2018, protocol no. 6-3/6). No financial support was received for the study.

## 3. Results

### 3.1. CD Readmissions

The total number of CD readmissions increased from 2014 to 2017 and then remained stable until 2020, when a significant drop was observed (Table 1). The mean age steadily increased over time from 39.7 ± 18.9 in 2014 to 48.6 ± 23.2 years, with a rapid peak in 2020, where the mean hospitalization age was 60.4 ± 22.6 years. Female patients prevailed throughout the study period and were significantly older than male patients; the gender gap varied from 14.9 to 32.1 years. There is a statistically significant difference in the age between gender (*p* < 0.001), where women are on average 26.1 years [95% CI 19.3–32.8] older than men. There is a statistically significant difference in the gender ratio (*p* = 0.008), where women are on average 16% [95% CI 5–27%] more common than men. The majority of patients were readmitted to university hospitals with gastrointestinal diagnosis; however, the number of gastrointestinal surgeries remained flat throughout the study period.

### 3.2. UC Readmissions

The total number of UC readmissions doubled from 2014 to 2018, with a significant drop in 2020 (Table 2). The mean age was stable over time, varying from 53.5 ± 20.5 to 62.0 ± 16.9 years. Female patients prevailed through the study period and were significantly older than the male patients. There is a statistically significant difference in the gender ratio (*p* = 0.011), where women are on average 18% [95% CI 5–31%] more common than men. There is a statistically significant difference in the age between gender (*p* = 0.039), where women are on average 5.9 years [95% CI 0.4–11.4] older than men. Almost half of the patients were readmitted to university hospitals every year. The number of surgeries was relatively stable over time, with a declining trend from 2015 to 2018.

There was no trend for the 30 d all-cause readmission rate of CD; however, for the CD-specific readmission rate, there was a statistically significant increasing trend, with an increase from 9.2% in 2014 to 15.7% in 2018 (*p*-trend = 0.0030) (Table 3). There was a statistically significant trend for the 30-day all-cause readmission rate of UC (*p*-trend = 0.018) in the period from 2014 to 2019 (54 to 93 emergency readmissions); however, no statistically significant trend was observed for the UC-specific readmission (*p*-trend > 0.05). There was no trend for the mean LOS for both patient groups with CD and UC (*p*-trend > 0.05).

There was a prominent UC-specific disease readmission proportion difference compared to the CD-specific disease readmission proportion from 2014 to 2019 (Figure 1); however, there was no statistical significance (*p* > 0.05). The number of CD-specific admissions from 2014 to 2019 saw a statistically significant increasing trend (*p* trend = 0.013). The percentage of the CD all-cause disease readmission proportion and CD-specific disease readmission proportion remained stable and similar to the UC all-cause disease readmission proportion. In the 30-day any readmissions and 30-day IBD-specific rehospitalizations, no deaths were observed.

### 3.3. Exploratory Analysis—Treated vs. Untreated IBD Patients

In CD and UC patients exploratory analysis no differences were observed in the all-cause and IBD-specific 30-day emergency readmissions between treated and not-treated patients (*p* > 0.05).

### 3.4. Hospitalization Cost Analysis

The majority of patients with IBD are hospitalized to university hospitals with no statistically significant trend towards increasing the total in-hospital days for the 2014–2020 time period (Table 4). UC patients, however, have more total in-hospital days also in city and regional hospitals, besides the university hospital being the common one. Mean total hospitalization costs (THC) in both CD and UC have no statistically significant changes over the 2014–2020 period. There is currently a five-level hospital system in Latvia. The highest level is the fifth—it includes university hospitals in Riga, which can provide complex medical services, as well as specialized medical institutions. The fourth-level hospitals are the large regional hospitals, while the third level hospitals are mainly attached to the city.

## 4. Discussion

Identifying early readmissions within 30 days for patients with IBD is crucial due to the potential correlation between the quality of inpatient care, the heightened risks of adverse outcomes and the substantial strain they impose on the Latvian healthcare system in both financial terms and other resource allocation.

### 4.1. Biodemographic Characteristics and Hospitalization Trends in Crohn’s Disease Admissions

Research on hospitalization trends in CD admissions has shown a decrease in hospitalizations, particularly in younger patients, in recent years [12]. However, the prevalence of hospitalizations in CD remains high, with the exacerbation of the disease being a common cause [13]. Despite the introduction of new therapies, the number of hospitalizations for CD as the first-listed diagnosis has not significantly changed, and patient education initiatives such as disease awareness, early relapse identification and treatment adherence improvement strategies are needed to prevent hospitalizations [14]. In Brazil, variables such as perianal disease and stricturing or penetrating behavior have been associated with hospitalization in CD patients [15].

We can note an increasing trend in rehospitalization in older patients with CD (Table 1); the mean age increased from 49.0 ± 20.1 years to 54.6 ± 19.4 years over the 2014–2020 period. These findings may be explained by senescence in the population with CD and higher comorbidity risks. There are more female patients with CD being re-hospitalized than males over the 2024–2020 period (Table 1), which correlates with the literature [5]. We also observed a drop in total hospital days in 2020, which may be associated with the increased demand of hospital resources during the COVID-19 pandemic. At the same time, with the COVID-19 pandemic, we observe a numerical increase in emergency surgeries in UC patients, which might be an indicator of the delay in treatment and timely relapse management.

### 4.2. Crohn’s Disease Specific Readmission (Only Emergency Hospitalizations)

A range of studies have explored the hospitalization trends and characteristics of CD patients. Cohen et al. (2000) found that the majority of hospitalizations were due to surgery, with a significant cost burden [5]. Niv et al. (2020) reported that 23.3% of patients were hospitalized, with the goal of reducing hospitalization and surgery [13]. Malarcher et al. (2017) noted stable hospitalization rates despite new therapies, suggesting a need for patient education [14]. Frolkis et al. (2014) highlighted the high risk of postoperative complications and readmissions, particularly in older patients and those with comorbidities [16]. These findings underscore the need for targeted interventions to reduce hospitalizations and improve outcomes for CD patients. We also observed more emergency admissions in patients with CD over the 2014–2020 period, with an increasing tendency in length of stay (LOS). Although the total count of 30-day readmissions remained relatively consistent throughout this timeframe, there were notable peaks in readmissions observed in 2015, 2017 and 2018. There are age disparities between male and female groups, highlighting a noteworthy gap in age distribution among these populations (Table 2).

### 4.3. Biodemographic Characteristics and Hospitalization Trends for Ulcerative Colitis Disease Admissions

Research on hospitalization trends for ulcerative colitis (UC) patients has shown a decrease in hospitalizations, particularly in younger patients, following the introduction of biologic medications [12]. However, obesity has been associated with longer hospital stays and higher charges for UC-related admissions [17]. Predictors of hospital readmissions for UC include longer initial hospital stays, a lack of an endoscopy and depression [18]. Further studies are needed to identify factors that are predictive of hospitalization and surgery in UC patients [13]. In our study, similar rehospitalization patterns for UC and CD patients are observed. UC patients have more surgeries and surgeries associated with K diagnosis than the CD patient group, which may be associated with the hospitalization discharge diagnosis—K (diseases of the digestive system) and C (neoplasms) (Table 3). A systematic review by Fumery et al. (2018) found that 10–15% of patients with UC undergo colectomy within 5–10 years of diagnosis [19]. This is often due to chronic refractory disease, as highlighted by Baker et al. (2020) in a review of outcomes after elective surgery for UC [20]. The use of immunomodulators and anti-tumor necrosis factor medicines has increased over time, potentially impacting the need for surgery. However, the impact of these medical interventions on surgery rates and patient outcomes, including health-related quality of life, remains an area for further research [21]. Reimbursed biological therapy, including anti-TNFs, was introduced in Latvia in 2018 to a limited patient group with significant co-payment. Broader access and zero co-payment for biological therapy was achieved in 2019; therefore, to evaluate biological therapy impact on hospitalization rates in Latvia, additional studies are needed.

Despite the high readmission rate in Latvia, the surgical intervention for both CD and UC groups is smaller than observed in a fellow U.S. investigative study [22]. Additionally, there was no mortality observed in all IBD-specific readmissions in Latvian hospitals, both emergency and elective, indicating the good quality of care among IBD patients in Latvia. There is an increasing trend in 30-day readmissions overall; however, the IBD-specific readmission trend is stable, meaning there is an increasing role of comorbidities, mean patient age and other risk factors causing rehospitalization growth—such findings also correspond to the literature [13,23].

Research in Germany has shown an increase in hospitalization rates for CD over time. Despite the use of biologics, the number of surgeries for CD has remained stable [24]. A systematic review and meta-analysis found that hospitalization occurs in 23.3% of CD patients, with the goal of therapy being to prevent hospitalization and surgery [13]. In Sweden, the incidence of CD is high, with older age being a risk factor for surgery [25].

The exploratory analysis did not reveal any statistically significant differences between treated and not-treated IBD patients, which could be related to a small sample size (less than 10 patients in some subgroups) and a relatively high out-of-pocket market before 2018, where NHS-covered IBD therapy did not include biological therapy and patient co-payment was as high as 25–50%. It is probable that there is a substantial prevalence of untreated IBD in the community. Failing to prioritize care for patients with IBD can result in clinical harm or prolonged negative impacts on their quality of life [26].

We also calculated the IBD-specific hospitalization rate per 100,000 people for both CD and UC patient groups. The CD-specific hospitalization rate per 100,000 people trend is relatively stable over the investigation period being 10.4:100,000 in 2014 and 9.3:100,000 in 2020, and 33.2:100,000 in 2014 and 35.1:100,000 in 2020 for CD and UC, respectively. Comparing the calculated results with other European countries, the hospitalization rate in Latvia is lower in terms of CD hospitalization rates, whereas the UC hospitalization rates are similar to the Northern European region [27]; for example, England’s hospitalization rates for 2013 are 202.9 for CD and 149.5 for UC, respectively. Germany’s hospitalization rates per 100,000 in 2012 were 24.7 for CD and 18.9 for CD. The Czech Republic’s hospitalization rates per 100,000 in 2015 were 36.72 for CD and 17.11 for UC.

The increasing trend is statistically significant over the period 2014–2018; however, the trend interrupts in 2020, which can be associated with the COVID-19 global pandemic and related changes in admission flows where the gastroenterology capacity was reallocated to accommodate increasing numbers of COVID-19 patients. Alternatively, the ongoing COVID-19 global outbreak has the potential to significantly impact mortality rates within the prevalent IBD population. The Surveillance Epidemiology of Coronavirus Under Research Exclusion (SECURE-IBD) is an international registry that tracks patients with IBD who have tested positive for COVID-19. An updated, interactive online map of the SECURE-IBD registry is available in the related links section. Among individuals aged 60 years and above reported in SECURE-IBD, 20% experienced severe COVID-19 complications, such as ventilator use, admission to the intensive care unit, or death. Advanced age (adjusted odds ratio [OR] 1.04, 95% confidence interval [CI] 1.01–1.06) and the presence of two or more comorbidities (adjusted OR 2.9, 95% CI 1.1–1.78) were associated with adverse outcomes. Consequently, older individuals constituted the most vulnerable segment of the IBD population during the pandemic [28].

Hospitalization rates in the 21st century vary by the epidemiologic stage of each region; countries embedded in stage 3 (compounding prevalence) predominantly show stabilizing hospitalization rates, including those in North America and Northern Europe [23].

The increased hospitalization trends mirrored the increasing incidence of IBD observed in the previous article [29] and others reporting increasing hospitalization rates for IBD (The Netherlands, þ3.25% per year, Portugal, þ1.92% per year) [30]. Often, the first year after diagnosis is associated with the highest risk of hospitalization for IBD. Furthermore, barriers to accessing expensive medications such as biological therapies may lead to worse disease severity, which results in hospital-based management [31,32,33].

Findings from a European Epi-IBD cohort health expenditure study in IBD patients indicated mean cost per patient-year during follow-up was EUR 2609 (SD 7389; median EUR 446 [IQR 164–1849]). Hospitalizations and diagnostic procedures accounted for more than 50% of costs during the first year [34]. These results are in line with our study findings, where readmission costs varied from EUR 962 to EUR 1132.

## 5. Strengths and Limitations

The key strength of this study is the 7-year period, which helps to identify any existing trends in readmission patterns. This is the first study analyzing readmission patterns in IBD patients in Latvia. Additionally, it reflects close to 100% of the population of patients who are hospitalized in Latvia that have universal health coverage. However, important limitations exist with this study. The NHS database does not contain data on the severity of the disease and, therefore, we were unable to further stratify the readmissions based on the severity of CD or UC. The NHS also lacks data on the total duration of the illness and the exact duration after discharge to readmissions, limiting our ability to assess index admissions more prone to earlier readmissions. Despite these limitations, this study helps us better understand the hospitalizations characteristics and trends of 30 d readmissions for CD and UC, which is critical for management of these patients.

## 6. Conclusions

In conclusion, the total number of CD and UC readmissions increased, and the decrease in admission rate was observed in 2020. Female patients prevailed through the study period and were significantly older than the male patients in both the CD and UC groups. We noted that there was no trend for the 30 day all-cause readmission rate for CD; however, there was a statistically significant trend for 30 day all-cause readmission for UC patients. There was a statistically significant trend for CD-specific readmission rate; however, no statistically significant trend was observed for UC-specific readmission. Exploratory analysis did not reveal any statistically significant differences between treated and not-treated IBD patients. Future prospective studies would be beneficial for further investigation and a comprehensive overview.

## Figures and Tables

**Figure 1 medicina-60-01310-f001:**
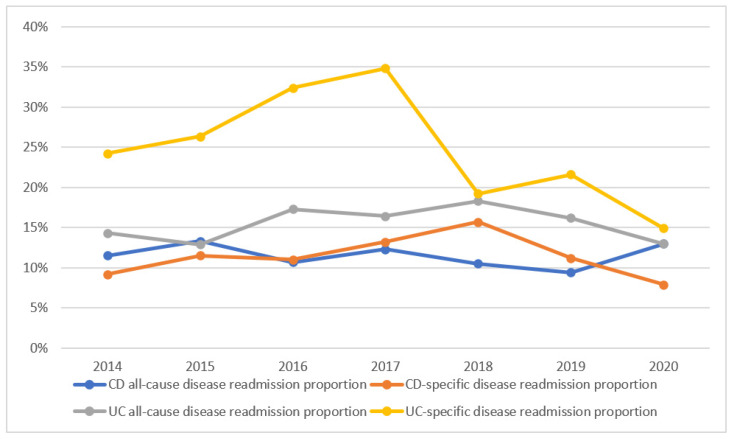
Trends of 30 day readmission following Crohn’s disease and ulcerative colitis hospitalizations. Abbreviations: CD—Crohn’s disease, UC—ulcerative colitis.

**Table 1 medicina-60-01310-t001:** Biodemographic characteristics and hospitalization trends for Crohn’s disease emergency 30-day readmission.

Variable	Year						
	2014	2015	2016	2017	2018	2019	2020
Number of emergency 30-day readmissions	24	34	27	32	29	25	23
Age (mean ± SD, years)	39.7 ± 18.9	42.9 ± 15.9	46.1 ± 22.0	44.6 ± 20.9	43.7 ± 21.1	48.6 ± 23.2	60.4 ± 22.6
**Gender (%)**							
Females	45.8%	55.9%	66.7%	56.3%	55.2%	52.0%	73.9%
Males	54.2%	44.1%	33.3%	43.8%	44.8%	48.0%	26.1%
Mean age females, years	47.8	51.3	55.4	58.6	56.9	63.6	67.7
Mean age males, years	32.9	32.2	27.6	26.5	27.5	32.4	39.8
Gender age gap, years	14.9	19.1	27.8	32.1	29.4	31.2	27.9
**Hospital type**							
City	8%	26%	33%	38%	31%	24%	26%
Regional	4%	12%	15%	22%	24%	40%	26%
University	88%	62%	52%	41%	45%	36%	48%
**Hospitalization discharge diagnosis**							
K	78.3%	78.6%	92.3%	75.0%	79.3%	76.0%	73.9%
A	4.3%	14.3%	3.8%	9.4%	0.0%	8.0%	8.7%
M	17.4%	7.1%	3.8%	6.3%	13.8%	8.0%	17.4%
C	0.0%	0.0%	0.0%	9.4%	6.9%	8.0%	0.0%
**Average LOS, days**							
City	11.0	5.4	11.0	7.9	6.8	7.8	5.4
Regional	5.0	8.3	5.0	20.8	6.0	10.8	14.7
University	5.2	11.5	8.6	21.7	14.6	5.0	10.2
Total in-hospital stay related to emergency 30-day readmission, days	104	258	196	379	293	174	163
Number of surgeries	1	4	1	1	1	1	1

Abbreviations: SD—standard deviation, yr—years, LOS—length of stay, A—infectious and parasitic diseases, K—diseases of the digestive system, C—neoplasms, M—diseases of the musculoskeletal system and connective tissue.

**Table 2 medicina-60-01310-t002:** Biodemographic characteristics and hospitalization trends for ulcerative colitis emergency 30-day readmission.

Variable	Year						
	2014	2015	2016	2017	2018	2019	2020
Number of emergency 30-day readmissions	54	52	79	72	102	93	63
Age (mean ± SD, years)	61.0 ± 18.9	57.2 ± 19.3	53.5 ± 20.5	56.7 ± 19.5	55.8 ± 19.6	59.6 ± 15.5	62.0 ± 16.9
**Gender (%)**							
Females	77.8%	63.5%	67.1%	45.8%	47.1%	54.8%	57.1%
Males	22.2%	36.5%	32.9%	54.2%	52.9%	45.2%	42.9%
Mean age females, years	62.0	61.1	56.8	51.8	59.3	62.6	67.1
Mean age males, years	57.6	50.5	46.8	60.9	52.7	55.8	55.1
Gender age gap, years	4.4	10.6	10	−9.1	6.6	6.8	12
**Hospital type**							
City	29.6%	36.5%	25.3%	37.5%	37.3%	45.2%	54.0%
Regional	25.9%	15.4%	22.8%	23.6%	22.5%	14.0%	14.3%
University	44.4%	48.1%	51.9%	38.9%	40.2%	40.9%	31.7%
**Hospitalization discharge diagnosis**							
K	60.4%	54.9%	61.5%	59.7%	60.8%	59.3%	59.0%
A	9.4%	11.8%	5.1%	8.3%	7.8%	9.9%	6.6%
M	26.4%	23.5%	25.6%	27.8%	26.5%	26.4%	29.5%
C	3.8%	9.8%	7.7%	4.2%	4.9%	4.4%	4.9%
**Average LOS, days**							
City	9.1	9.7	5.5	5.3	6.4	7.3	8.5
Regional	7.8	5.8	9.0	12.5	8.1	7.6	6.3
University	8.5	8.6	10.4	7.4	11.5	10.1	9.3
Total in-hospital stay related to emergency 30-day readmission, days	336	345	639	456	709	522	362
Number of surgeries	4	2	1	1	1	3	4

Abbreviations: SD—standard deviation, yr—years, LOS—length of stay, A—infectious and parasitic diseases, K—diseases of the digestive system, C—neoplasms, M—diseases of the musculoskeletal system and connective tissue.

**Table 3 medicina-60-01310-t003:** Readmission rates and healthcare burden for 30-day emergency readmissions of Crohn’s disease and ulcerative colitis.

Variable	Year							*p* Trend 2014–2019	*p* Trend 2014–2020
	2014	2015	2016	2017	2018	2019	2020		
**Crohn’s disease**									
Number of admissions	209	256	253	261	276	267	177	>0.05	>0.05
Number of CD-specific admissions	98	104	118	114	115	134	63	**0.013**	>0.05
CD-specific hospitalization rate per 100,000 person	10.4	12.9	12.8	13.4	14.3	13.9	9.3	>0.05	>0.05
Number of emergency admissions	142	143	139	145	131	133	109	>0.05	>0.05
Number of 30-day readmissions	46	55	56	61	83	61	34	>0.05	>0.05
Number of emergency 30-day readmissions	24	34	27	32	29	25	23	>0.05	>0.05
All-cause 30 day readmission rate, %	11.5%	13.3%	10.7%	12.3%	10.5%	9.4%	13.0%	>0.05	>0.05
Mean length of stay, days	5.5	9.6	9.3	15.8	10.1	8.3	9.6	>0.05	>0.05
**Ulcerative colitis**									
Number of admissions	665	734	823	843	851	829	670	>0.05	***p* = 0.044** (2014–2018) *p* > 0.05 (2018–2020)
Number of UC-specific admissions	179	196	245	238	221	223	133	>0.05	>0.05
UC-specific hospitalization rate per 100,000 person	33.2	37.0	41.8	43.2	44.0	43.2	35.1	>0.05	>0.05
Number of emergency admissions	434	446	493	510	508	470	395	>0.05	***p* = 0.044** (2014–2018) *p* > 0.05 (2018–2020)
Number of emergency 30-day readmissions	54	52	79	72	102	93	63	**0.018**	>0.05
All-cause 30day readmission rate, %	14.3%	12.9%	17.3%	16.4%	18.3%	16.2%	13.0%	>0.05	>0.05
IBD-specific emergency 30-day readmission	7	12	26	24	24	20	11	>0.05	>0.05
IBD-specific 30-day readmission proportion, %	24.2%	26.3%	32.4%	34.8%	19.2%	21.6%	14.9%	>0.05	>0.05
Mean length of stay, days	8.4	8.4	8.9	8.0	8.6	8.7	8.4	>0.05	>0.05

Abbreviations: CD—Crohn’s disease, UC—ulcerative colitis, IBD—inflammatory bowel disease. Bold *p* values are shown as significant.

**Table 4 medicina-60-01310-t004:** Hospitalization cost analysis for CD- and UC-related admissions.

Variable	Year						
	2014	2015	2016	2017	2018	2019	2020
**Crohn’s disease, total in-hospital days**							
City	25	52	76	79	54	56	68
Regional	60	91	33	193	78	80	44
University	617	676	706	598	768	687	446
IBD-related total hospital day costs, EUR	107,783	124,361	123,388	129,630	137,116	124,947	83,417
Mean hospitalization day cost per patient, EUR	1100	1196	1046	1137	1192	932	1324
**Ulcerative colitis, total in-hospital days**							
City	185	231	264	311	294	307	227
Regional	196	164	279	254	204	183	86
University	889	912	1150	1179	1008	992	727
IBD-related total hospital day costs, EUR	187,077	191,102	248,308	254,378	218,656	214,569	150,608
Mean hospitalization day cost per patient, EUR	1045	975	1014	1069	989	962	1132

Abbreviations: IBD—inflammatory bowel disease.

## Data Availability

Raw dataset is available upon request.

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
