# Peer review of "National 30-Day Readmission Trends in IBD 2014–2020—Are We Aiming for Improvement?"

_medicina, 2024, doi:10.3390/medicina60081310_

Round 1

Reviewer 1 Report

Comments and Suggestions for Authors

Dear Authors

This article reports on the trends in 30-day readmissions among patients with IBD in Latvia between 2014 – 2020, which is an interesting topic. However, some concerns in this article need to be addressed.

1.      Please add the background before study aim in the abstract.

2.      Please provide a definition of “ICD” in the abstract. 

3.      Please replace “Gastroenterology” with “gastroenterology” in line 31.

4.      Please update “Covid-19” to “COVID-19”. 

5.      Please include the inclusion and exclusion criteria in the method section.

6.      Please make reference to Table 1 in the text.

7.      Please elucidate the meanings of the abbreviations below each Table.

8.      Please bold the significant P value in all Tables.

9.      Please add “Bold P values are shown as significant.” below all tables.

10.  Please remove (author, year) in lines 181, 183, 186, and 188.

11.  In the first citation, please place the full name before the word abbreviation, followed by the abbreviation (e.g. CD, etc.). Please correct.

12.  Please add et al. or co-workers after the first author in the discussion section.

13.  Please include the “Funding” and “Ethical approval” sections.

Comments on the Quality of English Language

Minor editing of English language required.

Author Response

Dear Editor,

Thank you for valuable comments, all are relevant and highly appreciated. All changes in the text are highlighted in yellow.

  1. Please add the background before study aim in the abstract.
    • Answer: Background information added according to the suggestion.
  2. Please provide a definition of “ICD” in the abstract. 
    • Answer: Thank you for suggestions, added.
  3. Please replace “Gastroenterology” with “gastroenterology” in line 31.
    • Answer: Corrected accordingly.
  4. Please update “Covid-19” to “COVID-19”. 
    • Answer: Corrected accordingly.
  5. Please include the inclusion and exclusion criteria in the method section.
    • Answer: Thank you for pointing this. We added exclusion criteria. Inclusion criteria are already specified in "Study population" section.
  6. Please make reference to Table 1 in the text.
    • Answer: Corrected accordingly.
  7. Please elucidate the meanings of the abbreviations below each Table.
    • Answer: Corrected accordingly.
  8. Please bold the significant P value in all Tables.
    • Answer: thank you for suggestion. Highlighted based on comment.
  9. Please add “Bold P values are shown as significant.” below all tables.
    • Answer: added according to the editors comment.
  10. Please remove (author, year) in lines 181, 183, 186, and 188.
    • Answer: removed per comment.
  11. In the first citation, please place the full name before the word abbreviation, followed by the abbreviation (e.g. CD, etc.). Please correct.
    • Answer: Thank you, corrected.
  12. Please add et al. or co-workers after the first author in the discussion section.
    • Answer: Thank you, corrected.
  13. Please include the “Funding” and “Ethical approval” sections.
    • Answer: Thank you, information added.

Additionally, we performed minor editing of English language to improve readability.

Reviewer 2 Report

Comments and Suggestions for Authors

Dear Authors,

I read with interest this paper about national 30-day readmission trends in IBD 2014-2020. The manuscript is interesting and well written and the argument is not so frequent in literature. English is well readable and fluent. 

However, there are a few points about which I have questions or comments:

-        Abstract: Please specify the acronyms (CD/UC) before the use.

-        Table 1: I would suggest to adjust table layout.

-        Discussion: Line 180: please add the acronym ‘’CD’’ instead of Crohn’s disease. The same comment also applies to lines: 189; 192, 193; 199; 206; 207…

-        Discussion: Please clarify or add a comment to the concept of ‘’patient education’’.

-        Plea

It would also be interesting to know if there are differences in the readmission trend in relation to the extent of the disease (for ulcerative colitis) or the pattern of the disease (for Crohn's disease).

Best regards.

Comments on the Quality of English Language

English is well readable and fluent. 

Author Response

Dear Editor,

Thank you for valuable comments, all are relevant and highly appreciated. All changes in the text are highlighted in yellow.

  1. Abstract: Please specify the acronyms (CD/UC) before the use.
    • Answer: Thank you for suggestions, added.
  2.  Table 1: I would suggest to adjust table layout.
    • Answer: Thank you for suggestion. Layout format adjusted to improve readability. Please kindly specify in case additional editing is needed. 
  3. Discussion: Line 180: please add the acronym ‘’CD’’ instead of Crohn’s disease. The same comment also applies to lines: 189; 192, 193; 199; 206; 207…
    • Answer: Corrected accordingly.
  4. Discussion: Please clarify or add a comment to the concept of ‘’patient education’’.
    • Answer: Thank you for pointing this. We added a comment.
  5. It would also be interesting to know if there are differences in the readmission trend in relation to the extent of the disease (for ulcerative colitis) or the pattern of the disease (for Crohn's disease).
    • Answer: Thank you for suggestions. NHS database primarily captures moment of service delivery and fact of government payment and is not designed to register extend or pattern of disease. Therefore it is not possible to include above mentioned parameters in the study without individual hospital health record patient level data analysis. Additional studies are needed to perform such analysis, or creation of national level comprehensive IBD registry.

Additionally, we performed minor editing of English language to improve readability.

Reviewer 3 Report

Comments and Suggestions for Authors

Dear authors, thank you for the opportunity to review this very well designed manuscript.

I have got just a few comments and suggestions:

*It is not clear to me why you have used codes such as infectious and parasitic diseases (ICD codes - A00-B99), Diseases of the digestive system 84 (K00-K93), neoplasms (C00-D48), and Diseases of the musculoskeletal system and connec- 85 tive tissue (M00-M99). Could you clarify that, please?! I think you should have used only IBD-specific readmissions.

*The cost analysis is a secondary objective, and this should be clarified in the introduction and methods. Additionally, it might be relevant to clearly state this in the abstract as well.

*You have only provided diagnosis codes for the readmissions. It would be helpful to specify the main causes of readmissions. 

*I think you should update the discussion section, citing this publication: Dharni K, Singh A, Sharma S, Midha V, Kaur K, Mahajan R, Dulai PS, Sood A. Trends of inflammatory bowel disease from the Global Burden of Disease Study (1990-2019). Indian J Gastroenterol. 2024 Feb;43(1):188-198. doi: 10.1007/s12664-023-01430-z. Epub 2023 Oct 3. PMID: 37783933.

Author Response

Dear Editor,

Thank you for valuable comments, all are relevant and highly appreciated. All changes in the text are highlighted in yellow.

  1. It is not clear to me why you have used codes such as infectious and parasitic diseases (ICD codes - A00-B99), Diseases of the digestive system 84 (K00-K93), neoplasms (C00-D48), and Diseases of the musculoskeletal system and connec- 85 tive tissue (M00-M99). Could you clarify that, please?! I think you should have used only IBD-specific readmissions.
    • Answer: Thank you for your question. For all patients identified as IBD (had at least 1 reimbursed medicine prescription for IBD) we extracted all hospital admissions for IBD and associated diseases (which we defined as "all-cause readmission", and includes ICD-codes  such as infectious and parasitic diseases (ICD codes - A00-B99), Diseases of the digestive system 84 (K00-K93), neoplasms (C00-D48), and Diseases of the musculoskeletal system and connective tissue (M00-M99)). Separately we analyzed IBD-specific readmission, where primary hospitalization cause was captured as K50.X (Crohns disease) and K51.X (Ulcerative colitis). Thus keeping both parameters ("all-cause readmission" and "IBD-specific" readmission) we broaden the scope of the study and provide additional insights on potential associated extra-intestinal manifestation burden.
  2. The cost analysis is a secondary objective, and this should be clarified in the introduction and methods. Additionally, it might be relevant to clearly state this in the abstract as well.
    • Thank you for pointing this! Information added in the abstract, introduction and methods. 
  3. You have only provided diagnosis codes for the readmissions. It would be helpful to specify the main causes of readmissions. 
    • Answer: Thank you! We structured the main causes of readmission into "all-cause" (which includes ICD codes - A00-B99, K00-K93, C00-D48, M00-M99) and "IBD-specific", covering only readmissions where primary hospitalization diagnosis at discharge was K50.X and K51.X.
  4. I think you should update the discussion section, citing this publication: Dharni K, Singh A, Sharma S, Midha V, Kaur K, Mahajan R, Dulai PS, Sood A. Trends of inflammatory bowel disease from the Global Burden of Disease Study (1990-2019). Indian J Gastroenterol. 2024 Feb;43(1):188-198. doi: 10.1007/s12664-023-01430-z. Epub 2023 Oct 3. PMID: 37783933.
    • Thank you for putting this research into our attention, very relevant to the context of discussion! Citation added.